# Extracellular Cold-Inducible RNA-Binding Protein: Progress from Discovery to Present

**DOI:** 10.3390/ijms26083524

**Published:** 2025-04-09

**Authors:** Monowar Aziz, Irshad H. Chaudry, Ping Wang

**Affiliations:** 1Center for Immunology and Inflammation, The Feinstein Institutes for Medical Research, Manhasset, NY 11030, USA; pwang@northwell.edu; 2Departments of Surgery and Molecular Medicine, Zucker School of Medicine, Manhasset, NY 11030, USA; 3Department of Surgery, University of Alabama at Birmingham, Birmingham, AL 35294, USA; ichaudry@uabmc.edu

**Keywords:** DAMPs, immunity, inflammation, immune cells, diseases

## Abstract

Extracellular cold-inducible RNA-binding protein (eCIRP) is a critical damage-associated molecular pattern (DAMP) that drives inflammation and tissue injury in hemorrhagic and septic shock, and has emerged as a promising therapeutic target. Since then, extensive research using preclinical models of diseases and patient materials has explored eCIRP’s role in driving inflammatory responses and its potential as a biomarker. The main objective of this comprehensive review is to provide a detailed overview of eCIRP, covering its discovery, role in disease pathophysiology, mechanisms of release and action, potential as a biomarker, and therapeutic strategies targeting eCIRP in preclinical models of inflammatory and ischemic diseases. We examine the molecular, cellular, and immunological mechanisms through which eCIRP contributes to disease progression, and explore both well-established and emerging areas of research. Furthermore, we discuss potential therapeutic strategies targeting eCIRP across a broad spectrum of inflammatory conditions, including shock, ischemia–reperfusion injury, neurodegenerative diseases, and radiation injury.

## 1. Introduction

Extracellular cold-inducible RNA-binding protein (eCIRP) is a critical and potent damage-associated molecular pattern (DAMP) with profound implications in a broad range of inflammatory diseases [1]. A relevant commentary by Peter Ward highlighted eCIRP as a novel mediator of inflammation in hemorrhagic and septic shock, and suggested it as a promising therapeutic target [2]. We dedicate this review to Dr. Ward, honoring his insightful commentary on our discovery of eCIRP and his dedicated and durable contributions to the field of sepsis research. While initially described as an intracellular RNA chaperone to protect against cold and other stressors [3], CIRP’s extracellular release during inflammatory conditions like trauma, hemorrhagic shock, ischemia/reperfusion (I/R) injury, stroke, and sepsis has redefined its biological significance [1]. This review provides an updated overview of eCIRP research since its discovery 12 years ago [4]. We will explore eCIRP’s release mechanisms, receptor interactions, and downstream inflammatory signaling cascades. Furthermore, we will delve into eCIRP’s diverse cellular effects, including its influence on immune cell activation, differentiation, and polarization, its impact on epithelial, endothelial, and neuronal cells, as well as its capacity to modulate processes like cell death, efferocytosis, phagocytosis, and trogocytosis [5,6,7,8,9]. These multifaceted actions are documented through in vitro studies, preclinical models, and clinical observations in human diseases. We will discuss eCIRP’s roles in specific pathological conditions, highlighting its potential as both a therapeutic target and a disease biomarker. We will also demonstrate emerging mechanistic insights and explore the potential roles of eCIRP in diseases beyond those currently characterized, offering perspectives on future research directions for this critical mediator of inflammation and disease. Despite several reviews highlighting its role in various disease states since its discovery [1,10,11], this novel DAMP warrants further discussion. Here, we summarize previous findings and provide an update on the most recent advances on eCIRP in inflammation.

## 2. Discovery

CIRP, a member of the cold shock protein family, is constitutively expressed at low levels in various tissues. CIRP was first identified in 1997 for its role in the cold-stress response of mouse fibroblasts [3]. Human CIRP, an 18–21 kDa polypeptide comprising 172 amino acids, is encoded by a gene located on chromosome 19p13.3. It contains an RNA-binding domain (RBD) and arginine-rich motif (RGG) (Figure 1), both of which contribute to its diverse cellular functions [12]. For more than 15 years following its discovery within cells, research on CIRP remained limited, with few publications focusing on its RNA chaperone function [3,12,13,14]. A pivotal shift occurred in 2013 with the groundbreaking discovery of extracellular CIRP (eCIRP) by one of the authors’ groups [4]. They hypothesized that CIRP, as a cold-shock protein, might be involved in the post-surgical (mainly due to prolonged operations or emergency surgeries) stress response, as patients can develop sepsis or septic shock, conditions often accompanied by hypothermia [15]. Surprisingly, their research revealed elevated systemic levels of cell-free CIRP in patients with hemorrhagic shock and sepsis [4]. Subsequent studies demonstrated that inflammatory and hypoxic stress induced CIRP expression and, importantly, its release into the extracellular space [4,16]. This finding dramatically expanded the scope of CIRP research. Previously confined to intracellular roles in non-immune cells, CIRP’s newfound extracellular presence implicated it in immune responses and a broader range of diseases. The discovery of eCIRP ignited a surge in research, with publications increasing dramatically to several hundred in a short period. This surge reflects the recognition of eCIRP as a novel DAMP capable of influencing cellular activation, exacerbating inflammation, and driving organ dysfunction and death. Furthermore, eCIRP has emerged as a potential biomarker for disease diagnosis, prognosis, severity, and mortality [4,17,18,19,20]. This review provides an update on the remarkable progress made in the twelve years since the initial discovery of eCIRP, focusing on the expanding understanding of its diverse roles in inflammation.

## 3. Secretion

CIRP’s de novo expression is induced by various stressors, including hypoxia, lipopolysaccharide (LPS), inflammatory mediators, and environmental, chemical, or radiation exposure [1,16,21]. This review summarizes the active and passive release mechanisms of eCIRP (Table 1). eCIRP is released by immune and non-immune cells in diverse inflammatory conditions, such as sepsis, hemorrhagic shock, I/R injury, stroke, and radiation injury [1,4,19,21]. The release of eCIRP upon LPS stimulation is relevant in the context of post-surgical bacterial infections, highlighting its role in the resulting inflammatory response. Similarly, beyond infectious conditions, tissues experiencing ischemia–reperfusion, hypovolemic shock, or stroke suffer from hypoxia, which induces CIRP expression and release. In both infectious and ischemic conditions, this increased CIRP expression is likely regulated by the transcription factors NF-κB and HIF-1α, which are activated during inflammation and hypoxia, respectively. The mechanisms of eCIRP release have been elucidated and confirmed using gene knockout strategies and pathway inhibitors. Lacking a signal peptide, eCIRP is unlikely to be released via the classical endoplasmic reticulum (ER)-Golgi pathway. Our pioneering work using GFP-tagged CIRP revealed that under hypoxic or inflammatory stimuli, CIRP translocates from the nucleus to cytoplasmic stress granules, which subsequently fuse with lysosomes [4]. Lysosomal exocytosis likely contributes to the active release of eCIRP, as evidenced by CIRP enrichment in the lysosomal compartment of hypoxic macrophages. Another example of active eCIRP release is mediated through exosomes upon stimulation of macrophages with LPS or under septic conditions [22]. CIRP interacts with CD63, a member of the tetraspanin family, often considered a cargo protein. This interaction leads to CIRP’s loading into exosomes intracellularly, which are then released by budding. Indeed, CIRP has been confirmed to be present on the exosome surface, and is thus functionally active, recognizing its receptors on target cells [11,22]. Indeed, the presence of eCIRP in exosomes was confirmed by demonstrating its absence in exosomes isolated from LPS-stimulated *Cirbp*^−/−^ mice. Passive release of eCIRP is primarily facilitated by cell death processes, including apoptosis, necrosis, pyroptosis, and necroptosis [4,16,23,24]. If apoptotic cells are not cleared by phagocytes, they undergo secondary necrosis, releasing intracellular components, including eCIRP. Our previous study suggests that inflammasome activation in macrophages, leading to gasdermin D (GSDMD) activation and pore formation, facilitates eCIRP release from pyroptotic cells, even in the absence of complete lysis [23]. CIRP’s small size (17 kDa) allows its passage through GSDMD pores. Release of eCIRP through the inflammasome-GSDMD pore was confirmed using *Gsdmd*^−/−^ mice or using GSDMD inhibitors glycine or disulfiram. Necroptosis, driven by receptor-interacting protein (RIP)1/3-kinase-dependent mixed lineage kinase domain-like (MLKL) activation, also releases eCIRP, particularly in macrophages stimulated by pro-inflammatory cytokines or LPS during sepsis [24]. Necrostatin-1 (Nec-1), a necroptosis inhibitor, prevented eCIRP release, confirming its release occurs via necroptosis. Extracellular traps (ETs) are nucleic acids, encompassing both vital (active) and suicidal (passive) processes [25]. Since CIRP is an RNA-binding protein, ETs (nucleic acids) may also contain CIRP and contribute to eCIRP release. Further investigation is warranted to explore the involvement of other cell death modalities in eCIRP release during inflammation.

## 4. Receptors

The potential interaction of eCIRP with its receptors was investigated using BIAcore Surface Plasmon Resonance system, computational modeling, and FRET assays, which provide quantitative and qualitative measures of interaction. Extracellular CIRP, initially identified as a Toll-like receptor 4 (TLR4)/MD2 ligand, interacts with several receptors involved in immune responses [4]. BIAcore analysis revealed that eCIRP binds to TLR4, MD2, and the TLR4/MD2 complex with apparent *K*_D_ values of 6.17 × 10^−7^ M, 3.02 × 10^−7^ M, and 2.39 × 10^−7^ M, respectively. As a DAMP, eCIRP, like other DAMPs, can engage multiple receptors to exert diverse immune functions. While eCIRP binds to TLR2 and receptor for advanced glycation end products (RAGE), these interactions do not appear to elicit functional responses [4]. A recent study identified triggering receptor expressed on myeloid cells-1 (TREM-1) as a novel eCIRP receptor, confirmed by in silico modeling, BIAcore, and FRET assays [26]. eCIRP binding to TREM-1 (*K*_D_ = 11.7 × 10^−8^ M) activates macrophages and neutrophils, triggering the release of pro-inflammatory mediators and amplifying inflammation [26,27,28]. Because TREM-1 potentiates TLR4 signaling [29], this interaction also enhances TLR4 signal transduction, optimizing the immune response by increasing pro-inflammatory cytokine production. This is supported by our findings showing significantly diminished eCIRP effects in TREM-1-deficient macrophages and mice [26]. This also pinpoints a novel aspect whether eCIRP is involved in multiple receptor interaction to form a complex signaling system for generating eCIRP’s optimum detrimental impacts. Furthermore, our group identified the IL-6 receptor (IL-6R) as an unconventional eCIRP receptor, confirmed by in silico modeling, BIAcore, and FRET assays [30]. eCIRP binding to IL-6R (*K*_D_ = 9.8 × 10^−8^ M) promotes immune tolerance, a phenomenon observed in sepsis and other inflammatory diseases. This tolerance can lead to detrimental outcomes by hindering pathogen clearance, highlighting the double-edged sword nature of DAMPs in disease pathophysiology. Exploring other innate immune receptors or sensors that interact with eCIRP to modulate innate immunity may reveal novel pathophysiological mechanisms and therapeutic targets for inflammatory diseases.

## 5. Signal-Transduction

We demonstrated eCIRP’s mechanisms by focusing on three aspects: the receptors it recognizes, the cells it interacts with, and the resulting outcomes/effects (Figure 2). eCIRP binds to the TLR4/MD2 complex, a receptor expressed on various cell types [4]. In macrophages and lymphocytes, eCIRP activates NF-κB through the TLR4/MD2 complex [4,31], increasing the expression of pro-inflammatory cytokines and chemokines, and T cell activation markers such as CD69 and CD25 [31]. eCIRP also promotes differentiation towards pro-inflammatory Th1-type T cells by activating the master transcription factor T-bet [31]. In neutrophils, eCIRP upregulates intercellular adhesion molecule-1 (ICAM-1) surface expression via the classical TLR4/MD2 and NF-κB pathways [32]. ICAM-1 then transduces downstream signals to activate peptidyl arginine deiminase 4 (PAD4)-dependent NET formation [27]. Single-cell RNA sequencing recently revealed eCIRP’s impact on neutrophil heterogeneity, identifying a novel subtype exhibiting both antigen-presenting and aged phenotypes [33]. This subtype, which we termed antigen-presenting aged neutrophils (APANs), appears to exaggerate CD4 T cell activation, ultimately priming neutrophils for NET formation. The generation of APANs is induced by eCIRP/TLR4. In the regulatory B cell population (B-1a cells), eCIRP promotes a shift from an anti-inflammatory to a pro-inflammatory phenotype [34]. This shift is characterized by increased TNFα and IL-6 production, decreased IL-10 production, and reduced surface expression of the immunoregulatory receptor sialic acid binding Ig-like lectin G (Siglec-G). In dendritic cells (DCs), a CIRP fusion protein (SIIN-CIRP), induces DC maturation, cytokine production, and migration in a TLR4-dependent manner [35]. SIIN-CIRP also enhances antigen presentation. In lung endothelial cells (ECs), eCIRP increases ICAM-1 expression through TLR4/MD2 and NF-κB pathways [5].

eCIRP binding to TLR4 not only induces pro-inflammatory cytokine production, cellular differentiation, and plasticity, but also triggers various cell death processes, including pyroptosis [23], necroptosis [24], and ferroptosis [36] in immune-reactive cells. Since pyroptosis induction requires the inflammasome pathway, eCIRP has been shown to induce NLRP3 inflammasome, caspase-1, and GSDMD activation in macrophages and other cell types [5,23]. A recent study, consistent with this finding, revealed that eCIRP induces the unconventional formation of extracellular traps in macrophages via the caspase-1 and GSDMD pathway [37]. Recent studies also demonstrate that eCIRP causes mitochondrial dysfunction, leading to mitochondrial DNA degradation in macrophages and endothelial cells [38]. This has been linked to the activation of the intracellular DNA sensor cyclic GMP-AMP synthase-stimulator of interferon genes (cGAS-STING) pathway, inducing type I interferon expression by macrophages—a novel pathophysiological mechanism in hemorrhagic shock-induced inflammation [39]. Recent studies have shown that pulmonary endothelial cell death in sepsis occurs via eCIRP-mediated PANoptosis [40], a novel cell death pathway involving the simultaneous occurrence of pyroptosis, apoptosis, and necroptosis [41]. This PANoptosis is mediated by increased expression of Z-DNA-binding protein 1 (ZBP1), induced by the sensing of intracellular mitochondrial DNA [42,43].

Following the discovery of TREM-1 as another eCIRP receptor, several studies have investigated its intracellular signal transduction and downstream effects. Upon binding to TREM-1, eCIRP activates the adaptor protein DAP12, followed by Syk activation, ultimately leading to the activation of the inflammatory transcription factor NF-κB [26]. Furthermore, eCIRP has been shown to induce inflammation in alveolar type II (ATII) cells via TREM-1, implicating a novel pathophysiological mechanism in eCIRP-induced acute lung injury (ALI) [44]. TREM-1-mediated eCIRP signaling has been implicated in inducing inflammation in macrophages [26], and NET formation by neutrophils [27].

Finally, eCIRP interaction with IL-6R, another receptor of eCIRP, promotes signal transducer and activator of transcription 3 (STAT3) activation, leading to increased expression of inhibitory co-receptors on macrophages and lymphocytes, inducing tolerance [30]. IL-6R-mediated impairment of bacterial phagocytosis has been demonstrated through a novel downstream pathway involving the formation of a complex between eCIRP-activated STAT3 and βPIX, preventing βPIX from activating Rac1, and thus promoting macrophage phagocytic dysfunction [8]. This novel mechanism of eCIRP-induced macrophage phagocytic dysfunction provides a new therapeutic target for ameliorating sepsis. Given eCIRP’s interaction with TLR4, TREM-1, and IL-6R, deeper insights into its downstream signal transduction pathways can be gained using multi-omic, metabolomic, chromatin, and transcriptional studies at single-cell resolution.

## 6. Pathophysiology

Extracellular CIRP, a prominent DAMP, plays a significant role in the pathophysiology of various diseases, as demonstrated in preclinical models and clinical studies. This summary focuses on eCIRP’s involvement in acute inflammatory conditions such as sepsis, shock, and I/R injury, given the extensive existing literature on this topic. Initial research on eCIRP, using both preclinical hemorrhagic shock and sepsis models and patient samples, revealed elevated serum eCIRP levels in both mice and critically ill/septic patients [4]. These elevated eCIRP levels, observed in the context of these inflammatory diseases, induce macrophage production of TNF-α, IL-1β, and IL-6, contributing to multi-organ dysfunction, including ALI and acute kidney injury (AKI) in sepsis [1,45,46]. Subsequent studies broadened the understanding of eCIRP’s role in other inflammatory conditions arising from intestinal, hepatic, and renal I/R, as well as ischemic stroke [47,48,49,50]. Preclinical studies using CIRP-deficient mice, along with clinical data analysis employing regression techniques to correlate eCIRP levels with disease severity, have provided further insights into the role of eCIRP in these pathological conditions. Investigations at the cellular and molecular levels have examined eCIRP’s effects on innate and adaptive immune cells, endothelial and epithelial cells, focusing on cellular activation, differentiation, cytokine production, phagocytic function, and the initiation of cell death pathways.

Emerging research has begun to explore the role of eCIRP in other, less understood areas of pathophysiology, notably radiation injury [21,51]. These studies have shown a significant increase in eCIRP levels following radiation exposure, both in vitro (cell culture supernatants of peritoneal macrophages) and in vivo (peritoneal lavage). Similarly, radiation exposure also upregulates TREM-1 expression on peritoneal macrophages in both in vitro and in vivo settings [21]. The importance of both eCIRP and TREM-1 in radiation-induced pathology is highlighted by the improved survival rates observed in CIRP- and TREM-1-deficient mice following radiation exposure. Mechanistically, studies suggest that radiation-induced eCIRP impairs macrophage bacterial phagocytosis [51]. Radiation increases eCIRP release and, through its interaction with IL-6R, leads to downstream signaling (involving STAT3 activation) that impairs actin cytoskeletal rearrangements via the downregulation of Rac-1 and ARP2/3 [8,51]. This impaired clearance of bacteria is significant because it can increase the risk of sepsis following radiation injury, a consequence often associated with bacterial translocation from the intestine. Further research is needed to fully elucidate the role of eCIRP in these contexts.

Recent research suggests a role for eCIRP in neurodegenerative diseases, particularly Alzheimer’s disease (AD) [10,52]. Elevated eCIRP levels are observed in AD patients and are implicated in alcohol-induced memory impairment, suggesting a potential link between alcohol use, eCIRP, and AD progression [10,53,54,55,56]. Therefore, these studies demonstrate that alcohol increases eCIRP levels in the brain or blood, and that eCIRP promotes AD pathophysiology. Patient history, demographics, and clinical data may elucidate other factors contributing to AD development. In AD patients, eCIRP levels in cerebrospinal fluid (CSF) and plasma were significantly higher than those in age- and sex-matched control subjects [56]. Plasma eCIRP levels strongly correlated with levels of GFAP, an astrocyte activation marker, in AD patients [56]. Studies indicate that eCIRP promotes neuroinflammation via the neuronal IL-6Rα/STAT3/Cdk5 pathway and triggers Ca^2+^ release from the endoplasmic reticulum through an IL-6Rα/PLC/IP3-dependent mechanism, potentially contributing to Cdk5 activation and neurodegeneration [52,57]. CIRP may exacerbate AD pathophysiology through an alternative, intracellular mechanism. A study focusing on intracellular CIRP found a close association between CIRP and urokinase plasminogen activator (uPA), whose expression significantly decreased upon CIRP overexpression. Overexpression of CIRP in astrocytes inhibited uPA expression which, in turn, promoted Aβ_1–42_ production and tau phosphorylation in neurons, thereby increasing AD risk [55]. These results suggest that astrocytic CIRP overexpression contributes to AD development. While these findings implicate eCIRP in AD, further research is needed to fully elucidate its direct impact on tau pathophysiology and aggregation, the hallmarks of AD.

Pulmonary fibrosis (PF), a devastating complication of chronic inflammatory diseases, is characterized by progressive lung function decline and high mortality [58]. eCIRP promotes PF by inducing pro-inflammatory cytokines and activating lung fibroblasts via a TLR4/MD2/Myd88-dependent pathway [45,59,60]. In a bleomycin-induced mouse model of PF, eCIRP exacerbated disease and increased molecular markers of fibrosis, indicating its key role in PF pathogenesis [59].

The role of CIRP in cancer has been studied on a limited scale, and the distinct impacts of intracellular versus extracellular CIRP remain poorly defined [61,62]. Recent studies have begun to elucidate the mechanistic link between CIRP’s inflammatory function and its oncogenic potential. In a mouse model of colitis-associated cancer (CAC), CIRP promoted increased TNFα and IL-23 expression in inflammatory cells [61]. CIRP knockout mice exhibited reduced susceptibility to CAC development and decreased expression of chronic inflammation markers (TNFα and IL-23) and anti-apoptotic proteins (Bcl-2 and Bcl-XL) in colonic lamina propria cells. Given that eCIRP is a major driver of NET formation [27], and NETs are known to contribute to tumor metastasis [63], further investigation into the direct role of eCIRP in cancer biology is warranted.

Published quantitative data on eCIRP levels in sepsis and Alzheimer’s disease patients are currently limited to small pilot studies with few samples. While the reported fold-change increases compared to healthy controls are preliminary, a study of 69 septic patients revealed significantly higher plasma CIRP levels in nonsurvivors than survivors (median 4.99 ng/mL vs. 1.68 ng/mL), correlating with APACHE II and SOFA scores [17]. Analysis of CSF and plasma from 12 and 37 Alzheimer’s patients, respectively, revealed eCIRP levels of approximately 2.5 ng/mL and 1 ng/mL, compared to 1.8 ng/mL and 0.2 ng/mL in healthy controls [56]. However, large-scale, multicenter studies are needed to validate these findings, establish eCIRP as a marker of disease severity, and enable comparisons across diverse clinical contexts. Given that uncontrolled inflammation is central to the pathogenesis of many diseases, further investigation into the role of eCIRP as a key driver of these inflammatory processes is warranted. A deeper understanding of eCIRP’s diverse impacts may reveal novel therapeutic targets.

## 7. Therapeutics

Since its initial identification as a DAMP in a rat hemorrhagic shock model in 2013 [4], various therapeutic strategies targeting eCIRP have been developed and evaluated preclinically. The initial studies using an anti-CIRP antibody in rats demonstrated reduced cytokine levels, organ injury markers, and improved survival [4]. Similar results were observed in CIRP knockout mice subjected to hemorrhagic shock and in septic mice treated with the anti-CIRP antibody [4]. These promising findings spurred the development of additional eCIRP-targeted therapeutics.

Given CIRP’s function as an RNA-binding protein, therapeutic strategies employing miRNAs and miRNA mimics have been explored. For instance, miR-130-3p binds eCIRP, inhibiting its ability to increase cytokine expression via TLR4 activation [18]. A modified, more stable version of this mimic, incorporating phosphorothioate O-methyl groups, has also been developed [64]. Additionally, a modified poly-adenosine (A) tail (A12) has shown affinity for eCIRP [65]. These three molecules (miR-130-3p, modified miR-130-3p, and A12) have demonstrated efficacy in reducing cell death, cytokine expression, and improving survival in mouse models of sepsis and I/R injury. Similarly, X-aptamers, chemically modified DNA aptamers that bind target molecules with high affinity and specificity, have shown promise in targeting eCIRP [66]. An X-aptamer identified through a bead-based library screen specifically binds CIRP, blocking its interaction with TLR4. This aptamer reduced CIRP-induced pancreatic acinar cell injury in vitro and L-arginine-induced pancreatic injury and inflammation in vivo, suggesting that X-aptamers represent a promising therapeutic avenue for acute pancreatitis [66].

Scavenger molecules offer a unique approach to DAMP-targeted therapy. One such molecule, milk fat globule-epidermal growth factor VIII (MFG-E8), is a naturally occurring glycoprotein that scavenges apoptotic cells and also binds eCIRP [67]. However, its size and complexity hinder therapeutic development. Therefore, a smaller, high-affinity derivative peptide, MFG-E8-derived oligopeptide 3 (MOP3), was created. MOP3 binds eCIRP and then interacts with α_v_β_3_ integrin via its RGD sequence, facilitating phagocytosis by macrophages and intestinal epithelial cells [67,68]. This action reduces injury and improves survival in preclinical models of sepsis, neonatal sepsis, necrotizing enterocolitis, and I/R injury [67,68,69,70].

C23, a peptide derived from the eCIRP region that binds TLR4, exhibits higher affinity for the MD2 component of the TLR4/MD2 receptor complex than eCIRP itself [4]. This enhanced affinity translates to reduced inflammation and injury in mouse models of sepsis, hemorrhagic shock, and mesenteric I/R injury [1,4]. Similarly, M3, a peptide derived from the eCIRP region that binds TREM-1, has demonstrated therapeutic potential [26]. Studies in sepsis models revealed that TREM-1 knockout mice express lower levels of IL-6, TNFα, and IL-1β compared to wild-type mice after eCIRP injection [26]. Treatment with M3 attenuates cytokine expression, lung injury, and mortality in preclinical model of hemorrhagic shock [71]. Furthermore, M3 also mitigated eCIRP-mediated damage in a mouse model of mesenteric I/R injury [72].

A novel compound, opsonic peptide 18 (OP18), has recently been developed based on the premise that multiple DAMPs, including eCIRP, bind to TLR4 [unpublished data]. OP18 consists of an 18-amino acid (aa) peptide, comprising a 15-aa sequence targeting a binding site within the extracellular domain of TLR4 shared by eCIRP and other nuclear DAMPs, extended with an α_v_β_3_-integrin-binding RGD motif. This design allows OP18 to bind simultaneously to eCIRP and other DAMPs, while also interacting with α_v_β_3_-integrin on macrophages. This interaction promotes DAMP phagocytosis and lysosomal degradation. In preclinical sepsis models, OP18 has demonstrated efficacy in attenuating systemic inflammation and ALI, ultimately improving survival.

Although various neutralizing antibodies, peptides, miRNA antagonists, X-aptamer, and scavenging peptides have been effective in preclinical models, their bioavailability, side effects, and human testing remain to be thoroughly evaluated. The pharmacokinetics and half-life of the eCIRP’s antagonist A12 was studied in mice and cells [65], while emphasizing the need for further research to translate these findings to clinical practice. Research on eCIRP is expanding, with ongoing efforts to identify novel receptors, signal transduction pathways, and pathophysiological mechanisms. These discoveries are paving the way for the development of new therapeutics that target eCIRP to treat a broad spectrum of diseases in which it plays a critical role.

## 8. Perspectives

Extensive research has established eCIRP as a crucial DAMP, driving severe inflammation, organ damage, and mortality in various, primarily acute, inflammatory diseases. However, its role in chronic inflammatory diseases, which significantly impact global public health, remains less explored. Further investigation is needed to elucidate eCIRP’s involvement in conditions such as autoimmune diseases, hypertension, diabetes mellitus, ER stress, hypoxemia, hypoxia, cancer, and traumatic brain injury. While three prominent eCIRP receptors have been identified, its potential interactions with other molecules and receptors warrant further study. Currently, there are no studies demonstrating whether eCIRP interacts with LPS or flagellin, whether they share binding sites on TLR4, or if one influences the binding of the other. Similarly, no research has explored a direct interaction between eCIRP and the flagellin receptor, TLR5. Recognizing this gap, future investigations should examine eCIRP’s role in PRR pathobiology, including its potential interactions with other TLR ligands. Current research primarily focuses on eCIRP’s role in inducing gene expression, leaving its potential impact on gene repression largely unexplored. Similarly, understanding of eCIRP’s intracellular signaling, cellular activation, and heterogeneity is limited to conventional pathways. Multi-omic studies, including metabolic, chromatin, and transcriptomic analyses (e.g., single-cell ATAC-seq and RNA-seq using wild-type and CIRP^−/−^ mice), are needed to provide a more comprehensive view of eCIRP’s pathophysiology. This approach could reveal novel receptors and signaling pathways. Future research should also investigate eCIRP’s influence on chemotaxis, cell motility, and other cellular processes relevant to inflammatory diseases. Several promising eCIRP-targeting therapeutics have emerged from preclinical studies, and further evaluation of their pharmacokinetics, efficacy, and toxicity is necessary to advance them toward clinical trials.

## 9. Conclusions

In the twelve-year period since its discovery, eCIRP has been established as a critical inflammatory mediator in a wide range of diseases. This progress fuels hope that continued research on eCIRP will lead to the development of effective therapies targeting this molecule, offering potential cures for several life-threatening conditions.

## Figures and Tables

**Figure 1 ijms-26-03524-f001:**
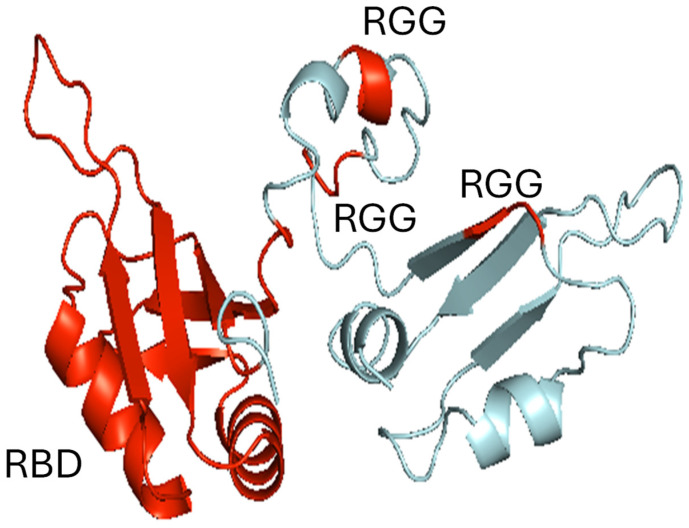
Three-dimensional configuration of CIRP. The human cold-inducible RNA-binding protein (CIRP; Q14011) amino acid sequence was retrieved from the UniProt database (https://www.uniprot.org/uniprotkb/Q14011/entry, accessed on 25 March 2025), and its protein structure was modeled using the I-TASSER tool (https://zhanggroup.org/I-TASSER/, accessed on 25 March 2025). I-TASSER generates structural models based on templates, using a threading approach to maximize percentage identity, sequence coverage, and confidence score. The resulting model was further refined, improving parameters such as the percentage of Ramachandran-favored residues and reducing the number of poor rotamers. Human CIRP contains an RNA-binding domain (RBD) (residues 6–84) and arginine–glycine–glycine (RGG)-rich domains (residues 94–96, 105–107, and 116–118). The RGG-rich domains are required for translational repression. The refined three-dimensional structural model was visualized using PyMOL v3.0.

**Figure 2 ijms-26-03524-f002:**
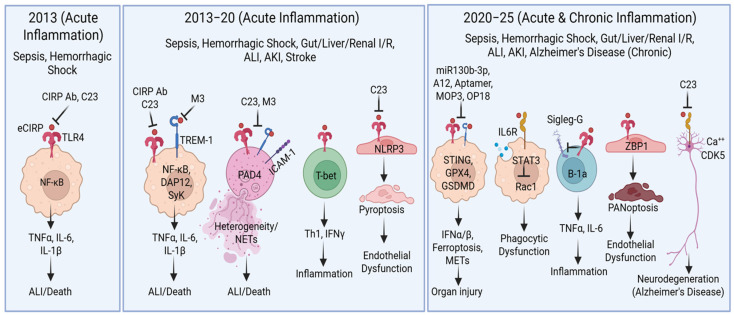
Role of extracellular CIRP in acute and chronic inflammatory diseases. eCIRP was discovered in 2013 as a critical DAMP that exacerbates inflammation and organ injury in sepsis and hemorrhagic shock via TLR4. From 2013 to 2020, considerable progress was made in elucidating its pathophysiology in a wide range of inflammatory diseases, including the discovery of a new receptor, TREM-1, and a new therapeutic compound targeting eCIRP/TREM-1 interactions. This period also depicts the discovery of eCIRP’s role beyond macrophages, in other cells such as neutrophils, T cells, and endothelial cells. Further progress in the eCIRP field occurred between 2020 and 2025, with the elucidation of another eCIRP receptor, IL-6R, which plays a pivotal role in immune tolerance. Additional discoveries included eCIRP’s role in macrophage phagocytic dysfunction, the polarization of B-1a cells from an immunoregulatory to a pro-inflammatory phenotype by targeting Siglec-G, the induction of endothelial cell PANoptosis by increasing ZBP1 expression, and the promotion of neurodegeneration by enhancing Ca^2+^ influx and CDK5 expression in neuronal cells, a novel pathophysiology in Alzheimer’s disease. eCIRP, extracellular CIRP; DAMP, damage-associated molecular pattern; TLR4, Toll-like receptor 4; TREM-1, triggering receptor expressed on myeloid cells-1; PANoptosis, pyroptosis, apoptosis, necroptosis; ZBP1, Z-DNA-binding protein 1; Siglec-G, sialic acid binding Ig-like lectin-G; CDK5, cyclin-dependent kinase 5.

**Table 1 ijms-26-03524-t001:** Pathways/mechanisms of eCIRP release.

Disease Conditions/ Stimulants	Release Pathways	Underlying Mechanisms	Cell Types	References
Sepsis/hemorrhagic shock, LPS, hypoxia, ER stress	Stress granule, lysosomal exocytosis	GSK3β, CKII	Macrophages	[4,13]
Sepsis, LPS, ATP, nigericin	Pyroptosis	Inflammasome, GSDMD	Macrophages	[23]
Sepsis, LPS, TNFα	Necroptosis	RIPK, MLKL	Macrophages	[24]
Sepsis, LPS	Exosomal	CD63 (Tetraspanin)	Macrophages	[11,22]

Abbreviations: LPS, lipopolysaccharides; ER, endoplasmic reticulum; GSK3β, glycogen synthase kinase-3 beta; CKII, casein kinase II; GSDMD, gasdermin D; RIPK, receptor-interacting protein kinase; MLKL, mixed lineage kinase domain-like; ATP, adenosine triphosphate; TNF, tumor necrosis factor.

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
