# Peer review of "Extracellular Cold-Inducible RNA-Binding Protein: Progress from Discovery to Present"

_ijms, 2025, doi:10.3390/ijms26083524_

Round 1
Reviewer 1 Report
Comments and Suggestions for Authors
In this paper of Aziz et al., the progress in the development of the research of eCIRP is summarized. The authors included a nice manuscript with nice figures illustrating the signaling mechanism upon binding of eCIRP, thus allowing the reader to easily follow the paper. I recommend the following points for improvement:
- I still do not get the main idea of this review. The authors need to clearly indicate their purpose of writing this manuscript.
- The abstract is just a repletion of the first sentences the authors used in the introduction. The authors need to rewrite their abstract.
- I have issue with consisting to describe the eCIRP as “new”, despite mention that it was discovered 12 years ago. May be the authors could describe this in other words. I do not want the readers to get an impression as if it was discovered in the last couple of years, which is not the case.
- The authors repeated a lot of information about the progress in the research and the paper published investigating eCIRP. Why is this important? I do not want the reads to be bored.
- The part related to the release of eCIRP is short and need to be expanded, since bacterial infections are common post-surgical complications.
- Could the authors provide some information about systems used for the secretion of eCIRP? In other words, which knockouts were used so far to study the secretion of eCIRP?
- Is this any relation between eCIRP and other ligands e.g. flagellin of bacteria, since they are an activating ligand of TLR-4?
- Please indicate which program was used to create the cartoons in this paper.
- I would recommend adding some diagrams e.g. 3D-diagrams illustrating the configuration of eCIRP.
Author Response
Reviewer-1:
In this paper of Aziz et al., the progress in the development of the research of eCIRP is summarized. The authors included a nice manuscript with nice figures illustrating the signaling mechanism upon binding of eCIRP, thus allowing the reader to easily follow the paper. I recommend the following points for improvement:
Response: Thank you for reviewing our manuscript and providing valuable feedback. We have addressed all your comments in a point-by-point manner.
- I still do not get the main idea of this review. The authors need to clearly indicate their purpose of writing this manuscript.
Response: We apologize for not clearly indicate the purpose of our work. The main objective of this comprehensive review is to provide a detailed overview of eCIRP, covering its discovery, role in disease pathophysiology, mechanisms of release and action, potential as a biomarker, and therapeutic strategies targeting eCIRP in preclinical models of inflammatory and ischemic diseases. We have explicitly stated this statement in the abstract, which has been extensively revised according to your suggestion to ensure clarity and alignment with the manuscript’s purpose (Line 16-21).
- The abstract is just a repletion of the first sentences the authors used in the introduction. The authors need to rewrite their abstract.
Response: We have revised the abstract to remove redundancy and ensure it presents a concise and standalone summary of the manuscript. The revised version now clearly outlines the main objectives and key findings without repeating sentences from the introduction (Line 9-26).
- I have issue with consisting to describe the eCIRP as “new”, despite mention that it was discovered 12 years ago. May be the authors could describe this in other words. I do not want the readers to get an impression as if it was discovered in the last couple of years, which is not the case.
Response: We appreciate your feedback. We have removed the term “new” to avoid any confusion regarding the timeline of eCIRP’s discovery and have instead characterized it as a critical DAMP (Line 30, 31).
- The authors repeated a lot of information about the progress in the research and the paper published investigating eCIRP. Why is this important? I do not want the reads to be bored.
Response: We appreciate your feedback on the redundancy issue. We have carefully revised the manuscript to streamline the discussion of research progress and published studies on eCIRP. Specifically, we removed repetitive details and reorganized the content so that each mention of research progress serves a clear purpose—either to provide essential context, highlight critical advancements, or support key arguments. These changes aim to maintain reader engagement and ensure that the narrative remains both informative and concise without overwhelming the reader with redundant information.
- The part related to the release of eCIRP is short and need to be expanded, since bacterial infections are common post-surgical complications.
Response: Thank you for the valuable suggestion. We have expanded the section on the release of eCIRP to include a more detailed discussion on its relevance to post-surgical bacterial infections. Specifically, we have elaborated on how bacterial components, such as LPS, trigger eCIRP expression and release, highlighting its role in the inflammatory response observed in these complications. This revision clarifies the link between bacterial infection and eCIRP release, ensuring the manuscript provides a comprehensive and engaging explanation for readers (Line 91-97).
- Could the authors provide some information about systems used for the secretion of eCIRP? In other words, which knockouts were used so far to study the secretion of eCIRP?
Response: We have expanded the discussion on the secretion systems of eCIRP in the revised manuscript. Specifically, we now describe that eCIRP is released via both passive and active mechanisms. Passive release occurs during cell death and necrosis, whereas active release involves several mechanisms, including lysosomal secretion, exosome-mediated release, gasdermin D-mediated pore formation (pyroptosis), and necroptosis. We have included details of studies employing CIRP and gasdermin D knockout mice, which demonstrated exosome-mediated and gasdermin D pore-dependent release. In addition, we discuss experiments using pathway-specific inhibitors—such as inhibitors of exosome release, gasdermin D, and necroptosis—to confirm the involvement of these mechanisms. These revisions, now incorporated on line 91-111, 118-120, 123-124, provide a comprehensive overview of the systems used to study eCIRP secretion.
- Is this any relation between eCIRP and other ligands e.g. flagellin of bacteria, since they are an activating ligand of TLR-4?
Response: We appreciate your insightful question regarding potential interactions between eCIRP and other TLR ligands such as flagellin. Currently, there are no studies demonstrating whether eCIRP interacts with LPS or flagellin, whether they share binding sites on TLR4, or if one influences the binding of the other. Similarly, no research has explored a direct interaction between eCIRP and the flagellin receptor, TLR5. Recognizing this gap, we have added a discussion in the Future Perspectives section (Line 390-395) to highlight the need for further investigation into eCIRP and PRR pathobiology, including its potential interactions with other TLR ligands.
- Please indicate which program was used to create the cartoons in this paper.
Response: We have updated the acknowledgments section to clearly state that the cartoons and figures in the paper were created using the BioRender software service (Line 416).
- I would recommend adding some diagrams e.g. 3D-diagrams illustrating the configuration of eCIRP.
Response: Thank you for your valuable suggestion. We have now incorporated a detailed 3D diagram as new Figure 1 to illustrate the structural domains and configuration of eCIRP. We have uploaded a new Figure 1 and 2 and their corresponding legends. We believe this addition along with other revisions significantly enhance the clarity, quality and visual appeal of our manuscript.
Reviewer 2 Report
Comments and Suggestions for Authors
The manuscript by Monowar Aziz, Irshad H. Chaudry and Ping Wang, Extracellular CIRP: the journey from discovery to the present day. The paper provides valuable insights on the role of eCIRP in diseases and potential treatments.
I propose to extend its clinical relevance beyond sepsis, especially by conducting more human Alzheimer's research. Further research into receptor interactions (TLR4, TREM-1, IL-6R) will improve understanding. Addressing treatment challenges such as bioavailability, side effects and testing in humans is important. More detailed information on macrophage eCIRP dysfunction in radiation-induced injury and more robust evidence of its role in Alzheimer's, particularly tau aggregation, is needed. In addition, the diagram would benefit from a clearer distinction between the roles of eCIRP in acute and chronic diseases, which would make the information more understandable. Similarly, the inclusion of quantitative data on eCIRP levels in different diseases (e.g. sepsis, cancer and Alzheimer's) in Table 1 would increase its value and usefulness.
Author Response
Reviewer 2:
The manuscript by Monowar Aziz, Irshad H. Chaudry and Ping Wang, Extracellular CIRP: the journey from discovery to the present day. The paper provides valuable insights on the role of eCIRP in diseases and potential treatments.
I propose to extend its clinical relevance beyond sepsis, especially by conducting more human Alzheimer's research. Further research into receptor interactions (TLR4, TREM-1, IL-6R) will improve understanding. Addressing treatment challenges such as bioavailability, side effects and testing in humans is important. More detailed information on macrophage eCIRP dysfunction in radiation-induced injury and more robust evidence of its role in Alzheimer's, particularly tau aggregation, is needed. In addition, the diagram would benefit from a clearer distinction between the roles of eCIRP in acute and chronic diseases, which would make the information more understandable. Similarly, the inclusion of quantitative data on eCIRP levels in different diseases (e.g. sepsis, cancer and Alzheimer's) in Table 1 would increase its value and usefulness.
Response: Thank you for your detailed and constructive feedback. We have addressed your suggestions as follows:
Clinical Relevance and Alzheimer's Research: We expanded the pathophysiology section to include a more in-depth discussion on eCIRP in human Alzheimer's disease (Line 275-281, 285-291). This addition provides further insight into its clinical relevance beyond sepsis.
Receptor Interactions: We have added detailed information on eCIRP’s interactions with receptors such as TLR4, TREM-1, and IL-6R. This now includes data on binding affinities (KD values) and the methodologies (e.g., Biacore, computational modeling, FRET assays) that have been used to characterize these interactions (Line 135-137, 139-141, 145, 146, 155, 156).
Therapeutic Challenges: We clarified the current status of therapeutic strategies targeting eCIRP. Although various neutralizing antibodies, peptides, miRNA antagonists, and scavenging peptides have been effective in preclinical models, their bioavailability, side effects, and human testing remain to be thoroughly evaluated. We have discussed the pharmacokinetics and half-life of the antagonist A12 studied in mice and cells, while emphasizing the need for further research to translate these findings to clinical practice (Line 373-377).
Macrophage Dysfunction and Radiation Injury: The manuscript now includes a more detailed account of eCIRP's role in macrophage dysfunction during radiation-induced injury. We describe how radiation increases eCIRP release and, through its interaction with IL-6R, leads to downstream signaling (involving STAT3 activation) that impairs actin cytoskeletal rearrangements via the downregulation of Rac-1 and ARP2/3 (Line 265-268). Additionally, we discuss its impact on bacterial phagocytic function.
Diagram Revision: We revised the diagram (now Figure 2) to clearly differentiate the cellular involvement and signaling pathways in acute (sepsis, ischemia-reperfusion, AKI, hemorrhagic shock, stroke) versus chronic (Alzheimer’s disease) inflammatory conditions. This should help readers better understand the distinct roles of eCIRP in various disease settings.
Quantitative Data Presentation: Published quantitative data on eCIRP levels in sepsis and Alzheimer's disease patients are currently limited to small pilot studies with few samples. While the reported fold-change increases compared to healthy controls are preliminary, a study of 69 septic patients revealed significantly higher plasma CIRP levels in nonsurvivors than survivors (median 4.99 ng/mL vs. 1.68 ng/mL), correlating with APACHE II and SOFA scores. Analysis of CSF and plasma from 12 and 37 Alzheimer's patients, respectively, revealed eCIRP levels of approximately 2.5 ng/mL and 1 ng/mL, compared to 1.8 ng/mL and 0.2 ng/mL in healthy controls. However, large-scale, multicenter studies are needed to validate these findings, establish eCIRP as a marker of disease severity, and enable comparisons across diverse clinical contexts (Line 310-319). Furthermore, the levels of eCIRP or its role in cancer development and progression remains largely unexplored, as only studied its role in colitis-associated cancer (CAC), an aspect we discuss in the perspective section.
We believe these revisions have significantly enhanced the manuscript’s clarity, clinical relevance, and overall impact.
Reviewer 3 Report
Comments and Suggestions for Authors
The review “Extracellular CIRP: Progress from Discovery to Present” by Aziz et al provides an interesting update of the various effects that have been attributed to this molecule. The main objective of this review is to provide an updated overview of eCIRP research since its discovery. The idea is interesting but does not consider various relevant aspects in order to establish various functions and participation in pathological processes, but they fail in trying to assign a functional value to this molecule. Otherwise, the presentation under these conditions is a summary of effects, without providing new information that would allow the readers of this journal to establish a critical analysis on this topic.
This molecule is a Cold-inducible RNA-binding protein (CIRP) belongs to the family of cold shock proteins which are constitutively but weakly expressed in various tissues and in particular play a relevant role in sterile inflammation, The human CIRP is an 18–21 kDa polypeptide containing 172 amino acids coded by a gene located on chromosome 19p13.3. CIRP has an RNA-recognition motif (RRM) and an arginine-rich motif (RGG), both of which have roles in coordinating numerous cellular activities. And various transcription factors such as hypoxia-inducible factor 1 alpha and nuclear factor-kappa B have been implicated in coordinating CIRP transcription in response to specific stimuli. , are details that I believe should be considered in the context of the work.
The authors present excessively the dates and some aspects related to the success or relevance of the discovery of this protein, it is perhaps of little relevance to promote interest in this topic e.g. abstract L 9 first discovered L 11The significance of this discovery was underscored by an accompanying commentary in the same journal, M; L 12 novel mediator; L13Since its discovery,; L 24, 26 and an extensive etcetera throughout the manuscript.
Author Response
Reviewer 3:
The review “Extracellular CIRP: Progress from Discovery to Present” by Aziz et al provides an interesting update of the various effects that have been attributed to this molecule. The main objective of this review is to provide an updated overview of eCIRP research since its discovery. The idea is interesting but does not consider various relevant aspects in order to establish various functions and participation in pathological processes, but they fail in trying to assign a functional value to this molecule. Otherwise, the presentation under these conditions is a summary of effects, without providing new information that would allow the readers of this journal to establish a critical analysis on this topic.
This molecule is a Cold-inducible RNA-binding protein (CIRP) belongs to the family of cold shock proteins which are constitutively but weakly expressed in various tissues and in particular play a relevant role in sterile inflammation, The human CIRP is an 18–21 kDa polypeptide containing 172 amino acids coded by a gene located on chromosome 19p13.3. CIRP has an RNA-recognition motif (RRM) and an arginine-rich motif (RGG), both of which have roles in coordinating numerous cellular activities. And various transcription factors such as hypoxia-inducible factor 1 alpha and nuclear factor-kappa B have been implicated in coordinating CIRP transcription in response to specific stimuli. , are details that I believe should be considered in the context of the work.
The authors present excessively the dates and some aspects related to the success or relevance of the discovery of this protein, it is perhaps of little relevance to promote interest in this topic e.g. abstract L 9 first discovered L 11The significance of this discovery was underscored by an accompanying commentary in the same journal, M; L 12 novel mediator; L13Since its discovery,; L 24, 26 and an extensive etcetera throughout the manuscript.
Response: We appreciate your detailed feedback. In response, we have revised the manuscript as follows:
Clarifying the Review’s Objective: We now emphasize that the primary aim of our review is to provide a comprehensive update on eCIRP research from its discovery to its current implications in pathophysiology, mechanisms of release and action, and therapeutic interventions (Abstract section, line 16-21).
Incorporating Detailed Molecular Information: In line with your suggestion, we have integrated detailed information on eCIRP’s structure and regulation. The revised manuscript now includes specifics on its 18–21 kDa size, 172 amino acid composition, RNA-binding domain (RBD), and arginine-rich motif (RGG) (Line 58-63), as well as the roles of transcription factors such as hypoxia-inducible factor 1α and nuclear factor-κB in its expression (Line 93-97).
Streamlining Historical and Success-Related References: We have eliminated repetitive mentions of discovery dates and accolades. Our timeline in Figure 1 has been refined to succinctly capture the major milestones without overemphasizing the historical context, thus maintaining reader interest and focusing on the molecule’s functional value. The text in lines 9, 11, 12, 13, and 24 has been amended accordingly. Line 26 has been revised to read: "A relevant commentary by Peter Ward highlighted eCIRP as a novel mediator of inflammation in hemorrhagic and septic shock, suggesting it as a promising therapeutic target", given that this review article was written for and submitted to the journal's special commemorative issue in honor of Professor Peter A. Ward: World Expert in Sepsis.
We believe that these revisions have adequately addressed your concerns and enhance the clarity and impact of our review.
Round 2
Reviewer 3 Report
Comments and Suggestions for Authors Unfortunately, the authors persist in maintaining a historical description of the discovery of this molecule, which detracts from the relevant data regarding its function and prospects for the study of this molecule. Although the modified version of the suggested structural features has included this information, unfortunately, it is not used to identify the function of this molecule. The work under these conditions does not provide novel information, and therefore its publication under these conditions is not recommended.Author Response
Unfortunately, the authors persist in maintaining a historical description of the discovery of this molecule, which detracts from the relevant data regarding its function and prospects for the study of this molecule. Although the modified version of the suggested structural features has included this information, unfortunately, it is not used to identify the function of this molecule. The work under these conditions does not provide novel information, and therefore its publication under these conditions is not recommended.
Response: Extracellular cold-inducible RNA-binding protein (eCIRP) was first identified by our group as a novel damage-associated molecular pattern (DAMP) mediating shock-induced injury, as reported in our 2013 Nature Medicine publication, which was accompanied by a commentary by Dr. Peter Ward in the same journal. We were invited by Professor Peter Ward to contribute a piece for submission to this Special Issue of the journal.
The comments suggest that the review does not present novel information or adequately discuss the functions of eCIRP. In response, we have clarified that, given our long-lasting expertise in this area, we have provided an in-depth discussion on the release, impact, mechanisms, pathophysiology, and therapeutic potential of eCIRP in various aspects of innate immunity. To our knowledge, this manuscript offers the most precise and comprehensive characterization of eCIRP as a DAMP, presenting insights that have not been described in previous reviews.
As the title of this work suggests, we cover eCIRP’s journey from discovery to its current progress, incorporating novel findings from literature published between 2013 and 2025. We believe this is the most up-to-date review on eCIRP, providing new perspectives on this molecule. We hope our response to your comments has addressed any concerns and that this manuscript will be considered for publication in honor of Dr. Peter Ward’s invitation, echoing his insightful commentary on this important molecule and our works.
Round 3
Reviewer 3 Report
Comments and Suggestions for Authors
t is too preliminar to be accepted
Author Response
Comment: After minor revisions, two reviewers accepted the manuscript, while the third one rejected it. This reviewer stated that the authors did not reply to the reviewer's concerns. The authors replied to the several concerns raised by the three reviewers.
Response: Thank you for your comment. In all review cycles, we responded to all comments from the reviewers, editors, and the editorial office in a point-by-point and timely manner. We appreciate your remark that "The authors replied to the several concerns raised by the three reviewers.
Comment: Looking at the literature, it is evident that the authors are the main group studying the extracellular CIRP. This is why the authors inserted several papers from their works and made a "historical" analysis of the topic. On the other hand, reviews on CIRP are present in the literature, and the authors' group is mentioned rarely. On the contrary, in this review, the authors, focusing on extracellular CIRP, not on CIRP complexively, cited their own work a lot. Anyway, they make the history of eCIRP.
Response: Thank you for your understanding that our group is the primary one studying extracellular CIRP (eCIRP). This focus led us to include several of our own publications and provide some historical context. However, we minimized the historical aspects in this paper to concentrate on the diverse facets of eCIRP, including its release mechanisms, actions, roles in disease pathophysiology, and novel therapeutic approaches targeting this molecule to alleviate inflammatory diseases. We wish to clarify that while several reviews exist on CIRP, only one focuses specifically on eCIRP, which we cited (Reference #1). Although CIRP and eCIRP represent different compartmentalized aspects of the same molecule, they have distinct pathophysiological and biological functions. This paper focuses solely on eCIRP.
Comment: It is unclear whether the reviewer who rejected the manuscript wanted a more comprehensive paper on eCIRP and CIRP.
Response: This comment is irrelevant and beyond our scope. However, we reiterate the distinction between eCIRP and CIRP. These represent two different aspects of the same molecule: eCIRP refers to its extracellular presence, where it acts as a damage-associated molecular pattern (DAMP) and is implicated in inflammation. CIRP, on the other hand, refers to its intracellular presence. Our expertise lies in eCIRP, not intracellular CIRP, and therefore this review focuses solely on the aspects of eCIRP.
Comment: It seems that the authors described well the relevance of the eCIRP in inflammation and sepsis.
It is unclear whether the complete absence of this specific form of CIRP in humans does not evoke an appropriate inflammatory response, and what the interrelationship is with other DAMPs and PAMPs. This point could be considered to further reinforce the idea of the role of this extracellular protein.
Response: Thank you for your understanding and positive comment regarding our description of eCIRP's relevance in inflammation and sepsis. CIRP is present in humans, and our Figure 1, which demonstrates the 3D configuration of CIRP, indeed depicts human CIRP. Human and mouse CIRP share 100% amino acid sequence homology. We have cited several papers (references 4, 10, 17, 18, 33, and 56) demonstrating the presence of eCIRP in humans and its inflammatory consequences in various inflammatory diseases. We hope this clarifies the point.
Comment: Figure 2 is not present in the manuscript, at least in the PDF. Please insert this figure.
Response: Figures 1 and 2, as well as the Table, are embedded in the uploaded manuscript.